

# `partR2`: partitioning R$^2$ in generalized linear mixed models

Martin A. Stoffel[1,2],  Shinichi Nakagawa[3] and  Holger Schielzeth[1]

[1] Institute of Ecology and Evolution, Friedrich-Schiller Universität Jena, Jena, Germany
[2] Institute of Evolutionary Biology, University of Edinburgh, Edinburgh, United Kingdom
[3] Evolution & Ecology Research Centre and School of Biological, Earth and Environmental Sciences, University of New South Wales, Sydney, Australia

## ABSTRACT

The coefficient of determination $R^2$ quantifies the amount of variance explained by regression coefficients in a linear model. It can be seen as the fixed-effects complement to the repeatability $R$ (intra-class correlation) for the variance explained by random effects and thus as a tool for variance decomposition. The $R^2$ of a model can be further partitioned into the variance explained by a particular predictor or a combination of predictors using semi-partial (part) $R^2$ and structure coefficients, but this is rarely done due to a lack of software implementing these statistics. Here, we introduce `partR2`, an R package that quantifies part $R^2$ for fixed effect predictors based on (generalized) linear mixed-effect model fits. The package iteratively removes predictors of interest from the model and monitors the change in the variance of the linear predictor. The difference to the full model gives a measure of the amount of variance explained uniquely by a particular predictor or a set of predictors. `partR2` also estimates structure coefficients as the correlation between a predictor and fitted values, which provide an estimate of the total contribution of a fixed effect to the overall prediction, independent of other predictors. Structure coefficients can be converted to the total variance explained by a predictor, here called 'inclusive' $R^2$, as the square of the structure coefficients times total $R^2$. Furthermore, the package reports beta weights (standardized regression coefficients). Finally, `partR2` implements parametric bootstrapping to quantify confidence intervals for each estimate. We illustrate the use of `partR2` with real example datasets for Gaussian and binomial GLMMs and discuss interactions, which pose a specific challenge for partitioning the explained variance among predictors.

# INTRODUCTION

Coefficients of determination $R^2$ are of interest in the study of ecology and evolution, because they quantify the amount of variation explained by a linear model (*Edwards et al., 2008*). By doing so, they go beyond significance testing in putting effects in perspective of the phenotypic variance. $R^2$ is expressed as a proportion of the total variance in the response, which represents a biologically relevant quantity if the total variation is representative for the total population (*De Villemereuil et al., 2018*). The total coefficient of determination in

Corresponding authors
Martin A. Stoffel,
martin.stoffel@ed.ac.uk
Holger Schielzeth,
holger.schielzeth@uni-jena.de

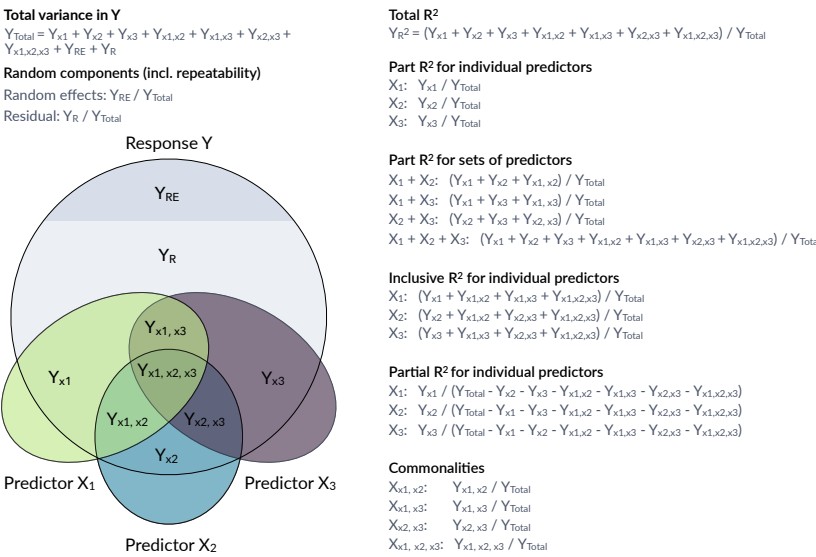

**Figure 1** **Conceptual framework for the estimation of proportions of variance components in a mixed model.** The large grey circle symbolizes the variance in a response Y, the dark grey area on the top indicates the share explained by random effects and the coloured ellipses symbolize variance in covariates with intersections indicating jointly explained variances. partR2 calculates total $R^2$, part $R^2$ for individual predictors and sets of predictors as well as inclusive $R^2$. The package does not report partial $R^2$ and commonalities, although they could be calculated from the partR2 output.

a generalised linear mixed model (GLMM) quantifies the variance explained by all fixed effects together (marginal $R^2$ *sensu Nakagawa & Schielzeth, 2013*, also known as the total correlation coefficient, *Watanabe, 1960*).

However, it is often of interest to attribute explained variation to individual predictors. Semi-partial coefficients of determination, also known as part $R^2$, decompose the variance of $R^2$ into components uniquely explained by individual predictors (*Jaeger et al., 2017*; *Jaeger, Edwards & Gurka, 2019*) or sets of predictors (Fig. 1). The set of all predictors in the model yields the total proportion of variance explained by the fixed part of the model (total $R^2$). With correlations among predictors, it often happens that predictors in univariate regressions explain a large share of the variance, but do not show large part $R^2$ if other correlated predictors are included in the model. Note that part $R^2$ estimates the proportion of the variance in the response explained by a predictor *while* accounting for covariance between this predictor and the other predictors in the model, whereas the (arguably more familiar) partial $R^2$ estimates the proportion of the variance that is explained by a predictor of interest *after* accounting for the other predictors from the response as well as the predictor of interest. The difference is subtle, but important (see more below). Therefore, part $R^2$ represents 'variance accounted for' in relation to the total variance, but partial $R^2$ does not. Consequently, part $R^2$ will be conceptually easier to compare with (total) $R^2$.

Structure coefficients provide a valuable addition to part $R^2$ in the decomposition of the phenotypic variance (*Nimon et al., 2008*; *Yeatts et al., 2017*). Structure coefficients quantify the correlation between individual predictors and the linear predictor. Predictors that

correlate well with a response, but are fitted with collinear predictors may show large structure coefficients as they are correlated to the predicted response, but low part $R^2$ as other predictors explain part of the same variance. Structure coefficients range from $-1$ to 1 with their absolute value expressing the correlation relative to a perfect correlation if a single predictor explains as much as the total fixed part of the model.

Structure coefficients are correlations and since the square of a correlation yields the variance explained, we can use structure coefficients to estimate the total variance explained by a predictor (*Nimon et al., 2008*). We call this the inclusive $R^2$ of a predictor and calculate it as the squared structure coefficient, i.e., its contribution to the linear predictor independent of other predictors (*Nimon et al., 2008*) times the proportion of variance explained by the linear predictor (which is the 'total' marginal $R^2$ of the model) (see also *Nathans, Oswald & Nimon, 2012*). As far as we are aware, inclusive $R^2$ has not been implemented before, but it provides valuable insights into the structure of the variance explained (Fig. 1).

Here, we introduce partR2, a versatile package for estimating part $R^2$, inclusive $R^2$, structure coefficients and beta weights from mixed-effects models. Figure 1 gives an overview of how variances are calculated and how they relate to partial $R^2$ and to commonality analysis (*Ray Mukherjee et al., 2014*; *Seibold & McPhee, 1979*; *Zientek & Thompson, 2006*). We illustrate how to use partR2 with real example datasets for Gaussian and binomial GLMMs, discuss how to estimate part $R^2$ in the presence of interactions and discuss some challenges and limitations. The Landesamt fuĺLr Natur, Umwelt und Verbraucherschutz Nordrhein-Westfalen "LANUV NRW" (Germany) approved this research (reference number: 84-02.04.2015.A439).

## MATHEMATICAL REPRESENTATION

### Part $R^2$

A Gaussian mixed-effects model can be written as:

$$\mathbf{y} = \mathbf{X}\boldsymbol{\beta} + \sum \boldsymbol{\alpha}_k + \boldsymbol{\varepsilon} \tag{1}$$

$$\alpha_k \sim N(0, \sigma^2_{\alpha_k})$$
$$\varepsilon \sim N(0, \sigma^2_{\varepsilon})$$

Where $\mathbf{y}$ is a vector of response values (outcomes), $\mathbf{X}$ is the design matrix of fixed effects, $\boldsymbol{\beta}$ is a vector of regression coefficients, $\sum \boldsymbol{\alpha}_k$ is the random part of the model that might contain multiple random effects and $\boldsymbol{\varepsilon}$ is a vector of residual deviations. The linear predictor $\beta$ represents the vector of predicted values from the fixed part of the model as $\beta = \mathbf{X}\boldsymbol{\beta}$. Note that we dealing with estimates of regression coefficients and variance components throughout (hence all $\boldsymbol{\beta}$ should be read as $\widehat{\boldsymbol{\beta}}$).

Since we are interested in the proportion of the phenotypic variance explained, we symbolize variance components by upper case $Y$ and index by the source of variance (as in Fig. 1). While variances are frequently represented as $V$ with the source of variance as an index, this leads to ambiguity for $V_X$ which might represent variance in $y$ explained by
$x$ or variance in $x$ itself, which is why we use this alternative notation. The total variance in the response is $Y_{Total} = \text{var}(y)$ and is estimated from the raw data or from the model (see below). The variance of the residuals is estimated by the model as $Y_R = \text{var}(\varepsilon)$. The variance of the (sum of) random effects is estimated by the model as $Y_{RE} = \text{var}(\sum \alpha_k)$ and the variance explained by fixed effects can be estimated as the variance in the linear predictor $Y_X = var(X\beta)$.

The coefficient of determination $R^2$ estimates the proportion of variance in the response that is explained by fixed effects. The coefficient of determination $R^2$ for the total fixed part of the model is thus:

$$R_X^2 = \frac{Y_X}{Y_X + Y_{RE} + Y_R} = \frac{Y_X}{Y_{Total}} \tag{2}$$

Note that the sum of the components in the denominator might deviate numerically from the total outcome variance in the raw data. However, conceptually they are the same in that they represent the population-level outcome variance. The variance in the outcome is an estimate from the specific sample, while the sum of components of the mixed model represents a population-level estimate given the data and the model.

A reduced model with a (set of) fixed effect predictors $X^*$ removed but the same random effect structure can be fitted as (now using the tilde to highlight the differences from Eq. (1)):

$$\mathbf{y} = \widetilde{\mathbf{X}}\widetilde{\boldsymbol{\beta}} + \sum \widetilde{\boldsymbol{\alpha}}_k + \widetilde{\boldsymbol{\varepsilon}} \tag{3}$$

$$\widetilde{\alpha}_k \sim N(0, \sigma_{\widetilde{\alpha}_k}^2)$$
$$\widetilde{\varepsilon} \sim N(0, \sigma_{\widetilde{\varepsilon}}^2)$$

with the variance in the linear predictor of the reduced model being $Y_{\widetilde{X}} = var(X\beta)$.

The variance uniquely explained by $X^*$ is then the difference between the variance explained by fixed effects in the full and the reduced model $Y_{X^*} = Y_X - Y_{\widetilde{X}}$. Part $R^2$ sets this variance in proportion to the total outcome variance:

$$R_{X^*}^2 = \frac{Y_X - Y_{\widetilde{X}}}{Y_X + Y_{RE} + Y_R} = \frac{Y_X - Y_{\widetilde{X}}}{Y_{Total}} \tag{4}$$

The process of fitting a reduced model, estimation of $Y_{X^*}$ and estimation of $R_{X^*}^2$ can be repeated for all predictors and combinations of predictors. At the limit for a model with all fixed effects removed, $R_{X^*}^2 = R_X^2$.

### Side-note on partial $R^2$

For completeness we note that the partial $R^2$ could be calculated as:

$$R_{X^*}^2 = \frac{Y_X - Y_{\widetilde{X}}}{Y_{Total} - Y_{\widetilde{X}}} \tag{5}$$

However, this estimate does not put the explained variance in perspective of the total variance in the response. It has the major disadvantage that the denominator depends on $Y_{\widetilde{X}}$. The same effect in terms of $Y_{X^*}$ thus appears larger if the reduced model explains more

variance (larger $Y_{\widetilde{X}}$). Even in the case of independent additive predictors, the contributions of the different fixed effects do not sum up to $R_X^2$, because of the change in the denominator that different $Y_{X*}$ are compared to. Finally, since we are interested in explaining phenotypic variation in some biological response (the phenomenon to be explained), we think that part $R^2$ is the more relevant quantity, as it represents the proportion of variance in the response uniquely explained by $X^*$.

## Inclusive $R^2$

Structure coefficients are the Pearson correlations between a particular predictor of interest $x^*$ and the linear predictor $\eta$. Note that we now use a lower case $x^*$ to indicate that we are dealing with a single predictor. Structure coefficients are quantified from the full model as:

$$SC_{x*} = \text{cor}(\eta, x^*) \tag{6}$$

The squared correlation between two variables $a$ and $b$ gives the variance explained for these variables $\text{cor}(a,b)^2 = R_a^2$. The squared structure correlations thus quantify the proportion of variance in the linear predictor $Y_X$ that is explained by a the predictor of interest $x^*$. Since the proportion of outcome variance explained by the linear predictor in the full model is $R_X^2$, the inclusive variance explained by predictor $x^*$ is:

$$IR_{x*}^2 = SC^2 \cdot R_{x*}^2 \tag{7}$$

Inclusive $R^2$ as we define it here, complements part $R^2$ by giving additional insights. While part $R^2$ quantifies the variance uniquely explained by a predictor (or set of predictors), inclusive $R^2$ quantifies the total proportion of variance explained in the model, both uniquely and jointly with other predictors. In the special case of a single predictor in a model $SC_{x*} = \text{cor}(\eta, x^*) = 1$, such that $IR_{x*}^2 = R_X^2$.

## Part $R^2$ in non-Gaussian models

For Gaussian models there is a single residual error term $\varepsilon$ with variance $Y_R = \text{var}(\varepsilon)$. For non-Gaussian models, however, there is additional error that arises from the link function that translates latent-level predictions to observed outcomes. This variance can be approximated for a variety of link functions and error distributions (*Nakagawa & Schielzeth, 2010*; *Nakagawa, Johnson & Schielzeth, 2017*). Our R package currently implements distribution-specific variances for Poisson models with log and square root link functions and binomial models with logit and probit link functions. For Poisson models and non-binary binomial models (proportion models), partR2 also fits an observational level random effect (if none is fitted already) to estimate variance due to overdispersion (*Harrison, 2014*). Both the overdispersion variance, now denoted $Y_R$ and the distribution-specific variance $Y_D$ are included in the denominator of the part R2 calculation:

$$R_{X*}^2 = \frac{Y_X - Y_{\widetilde{X}}}{Y_X + Y_{RE} + Y_D + Y_R} \tag{8}$$

Notably, there are other estimation methods for $R^2$ for non-Gaussian models or GLMM (*Jaeger et al., 2017*; *Piepho, 2019*). Currently, partR2 only implements the method based on *Nakagawa & Schielzeth (2013)* and *Nakagawa, Johnson & Schielzeth (2017)*.

## OTHER IMPLEMENTATIONS IN R PACKAGES

There are a few R packages that calculate part $R^2$ for linear models (lm), for example `rockchalk::getDeltaRsquare` (*Johnson & Grothendieck, 2019*). Other packages calculate partial $R^2$ (not part $R^2$) such as `asbio::partial.R2` (*Aho, 2020*) and `rr2::R2` (*Ives & Li, 2018*) for linear models and `rsq::rsq.partial` (*Zhang, 2020*) for linear models and generalized linear models (glm). Note that partial $R^2$ is different from part (semi-partial) $R^2$ (partial $R^2$ >part $R^2$), since it represents the unique variance explained by a particular predictor but after removing ('partialling out') the variance explained by the other predictors (*Yeatts et al., 2017*, Fig. 1). The ppcor package calculates semi-partial and partial correlations, but does not work on fitted GLM or GLMM models (*Kim, 2015*). The package yhat features functions for commonality analyses in glms (*Nimon, Oswald & Roberts, 2020*). None of these packages estimates part $R^2$ for mixed-effects models that we focus on here.

Several packages estimate (marginal) $R^2$ as the variance explained by all fixed effects in linear mixed-effects models. This includes `performance::r2_nakagawa` (*Lüdecke et al., 2020*), `MuMIn::r.squaredGLMM` (*Bartoń, 2019*), and `rptR::rpt` (*Stoffel, Nakagawa & Schielzeth, 2017*). These packages do not allow to estimate part $R^2$. The only versatile package to estimate part $R^2$ from linear mixed-models is r2glmm (*Jaeger, 2017*). The function `r2glmm::r2beta` computes part $R^2$ from lmer, lme and glmmPQL model fits (also for linear models lm and glm) based on Wald statistics. However, it does neither support `lme4::glmer` for generalized linear model fits nor does it allow to estimate $R^2$ for combinations of predictors. Furthermore, it does not estimate structure coefficients, inclusive $R^2$ or part $R^2$ for multilevel factors as a unit.

### Features of `partR2`

partR2 takes a fitted (generalized) linear mixed-model (GLMM), from the popular mixed model package lme4 (*Bates et al., 2015*) and estimates part $R^2$ by iteratively removing fixed effects (*Nimon et al., 2008*). The specific fixed effects of interest are specified by the `partvars` and/or by the `partbatch` argument. The package estimates part $R^2$ for all predictors specified in `partvars` individually and in all possible combinations (the maximum level of combinations can be set by the `max_level` argument). A custom specification of fixed effects of interest saves computation time as compared to an all-subset specification and is therefore required in partR2.

The central function partR2 will work for Gaussian, Poisson and binomial GLMMs. Since the model fit is done externally, there is no need to supply a family argument. For non-Gaussian GLMMs, the package estimates link-scale $R^2$ (*sensu Nakagawa & Schielzeth, 2013*). We implement parametric bootstrapping to quantify sampling variance and thus uncertainty in the estimates. Parametric bootstrapping works through repeated model fitting on simulated data based on fitted values (*Faraway, 2015*). The number of bootstrap iterations is controlled by the `nboot` argument. We recommend a low number of `nboot` for testing purposes and a large number (e.g., `nboot = 1000`) for the final analysis.

The package returns an object of class partR2 that contains elements for part $R^2$, inclusive $R^2$, structure coefficients, beta weights (standardized regression slopes), bootstrapping

iterations and some other information. An extended summary, that includes inclusive $R^2$, structure coefficients and beta weights can be viewed using the summary function. The forestplot function shows a graphical representation of the variance explained by individual predictors and sets of predictors along with their bootstrapping uncertainties. All computations can be parallelized across many cores based on the future and furrr packages (*Vaughan & Dancho, 2018*; *Bengtsson, 2020*). An extended vignette with details on the complete functionality accompanies the package.

## Example with Gaussian data

We use an example dataset with hormone data collected from a population of captive guinea pigs to illustrate the features of partR2. The dataset contains testosterone measurements of 31 male guinea pigs, each measured at 5 time points (age between 120 and 240 days at 30-day intervals). We analyze log-transformed testosterone titers and fit male identity as a random effect. As covariates the dataset contains the time point of measurement and a rank index derived from behavioral observations around the time of measurement (*Mutwill et al., 2021*).

*Rank* and *Time* are correlated in the dataset ($r = 0.40$), since young individuals are typically low rank, while older individuals tend to hold a high rank. *Time* might be fitted as a continuous predictor or as a factor with five levels. Here we present the version of a factorial predictor to illustrate the estimation of part $R^2$ for interactions terms. Hence, an interaction between *Time* and *Rank* will also be fitted.

First, the package needs to be loaded (after successful installation) in an R session (*R Core Team, 2021*). The package comes with the guinea pig dataset that also needs to be loaded using the data function.

```
library(partR2)

data(GuineaPigs)
```

A single record contains missing values for testosterone measurements. Missing records can be problematic to handle in partR2 and are better removed prior to the analysis. We also log-transform the response and convert *Time* to a factor and filter for the first three time points to simplify the output.

```
GuineaPigs <- subset(GuineaPigs, !is.na(Testo) & !is.na(Rank) & (Time
%in% c(1,3,5)))
GuineaPigs$TestoTrans <- log(GuineaPigs$Testo)
GuineaPigs$Time <- factor(GuineaPigs$Time)
```

We then fit a linear mixed effects model using lmer from the lme4 package (*Bates et al., 2015*). Further exploration of the data and model checks are omitted here for simplicity, but are advisable in real data analysis.

```
library(lme4)
mod <- lmer(TestoTrans ~Rank * Time + (1|MaleID), data=GuineaPigs)
```

The partR2 analysis takes the lmer model fit (an merMod object) and a character vector partvars indicating the fixed effects to be evaluated. Interactions are specified with the colon syntax (see the package's vignette for further details).

```
res <- partR2(mod, partvars = c("Rank", "Time", "Rank:Time"), nboot=100)
```
The function returns a `partR2` object. The `print` function reports the part coefficients of determination and a more extensive summary can be viewed with the `summary` function which also shows inclusive $R^2$, structure coefficients and beta weights (standardized slopes) (Fig. 2).

```
print(res)
summary(res, round_to = 2)
```
The variances appear largely additive, since combinations of predictors explain about the sum of the variance explained by individual predictors. The main components of the `partR2` object can be accessed for further processing as `res$R2` for part $R^2$ (with point estimates and confidence intervals), `res$SC` for structure coefficients, `res$IR2` for inclusive $R^2$ and `res$BW` for beta weights.

## Dealing with interactions

Models with interactions are problematic, because the variance explained by a main factor can be estimated in multiple ways (Fig. 3) and because of the internal parametrization of the model matrix.

The model output in Fig. 2 shows the number of parameters fitted in each model (each row in the R2 part refers to a reduced model). In the `print` and `summary` output this is visible as a column labelled 'ndf'. A close inspection shows that the removal of rank did not change the number of parameters (6 for the full model, 6 for the model excluding rank). This is because the model matrix is reparametrized in the reduced model and `lmer` will fit three terms for the interaction (here `Time1:Rank`, `Time3:Rank`, `Time5:Rank`) rather than just two for the interaction in the full model. Dummy coding of the factor can be usefully combined with centering of dummy coded variables (*Schielzeth, 2010*) and gives more control over this re-parametrisation. It allows for example to estimate the part $R^2$ for the average effect of *Rank* by constraining the average *Rank* effect to zero, so that only the two contrasts are fitted (here `Time3:Rank`, `Time5:Rank`):

```
GuineaPigs <- cbind(GuineaPigs, model.matrix(~0 + Time, data=GuineaPigs))
GuineaPigs$Time3 <- GuineaPigs$Time3 - mean(GuineaPigs$Time3)
GuineaPigs$Time5 <- GuineaPigs$Time5 - mean(GuineaPigs$Time5)
```
The model can then be fitted with dummy predictors. Since the usual specification in `partR2` via `partvars` would fit all possible combinations, including combinations of the different *Time* terms, such a run can take a long time. However we are mostly interested in fitting and removing all dummy predictors at a time. The package therefore features an additional argument `partbatch` to specify a list of character vectors containing the sets of predictors that should always be kept together. In the example, the list has two elements, a character vector for the dummy-coded main effects and a character vector for the interaction terms. The analysis yields part $R^2$ for two batches of predictors as well as *Rank* and their combinations.

```
mod <- lmer(TestoTrans ~(Time3 + Time5) * Rank + (1|MaleID),
data=GuineaPigs)
batch <- c("Time3", "Time5")
```

```
R2 (marginal) and 95% CI for the full model:
 R2   CI_lower CI_upper ndf
 0.17 0.09     0.36     6

──────────

Part (semi-partial) R2:
 Predictor(s)        R2   CI_lower CI_upper ndf
 Model               0.17 0.09     0.36     6
 Rank                0.00 0.00     0.18     6
 Time                0.02 0.00     0.20     4
 Rank:Time           0.04 0.00     0.21     4
 Rank+Time           0.02 0.00     0.20     4
 Rank+Rank:Time      0.16 0.08     0.34     3
 Time+Rank:Time      0.04 0.00     0.22     2
 Rank+Time+Rank:Time 0.17 0.09     0.36     1

──────────

Inclusive R2 (SC^2 * R2):
 Predictor  IR2  CI_lower CI_upper
 Rank       0.13 0.03     0.26
 Time3      0.00 0.00     0.04
 Time5      0.00 0.00     0.04
 Rank:Time3 0.05 0.01     0.13
 Rank:Time5 0.01 0.00     0.07

──────────

Structure coefficients r(Yhat,x):
 Predictor  SC    CI_lower CI_upper
 Rank       0.87  0.56     0.94
 Time3      0.14  −0.18    0.43
 Time5      0.16  −0.26    0.48
 Rank:Time3 0.56  0.22     0.75
 Rank:Time5 0.28  −0.14    0.57

──────────

Beta weights (standardised estimates)
 Predictor  BW    CI_lower CI_upper
 Rank        0.50 −0.08    0.94
 Time3      −0.19 −0.53    0.14
 Time5       0.17 −0.20    0.55
 Rank:Time3  0.17 −0.36    0.83
 Rank:Time5 −0.36 −0.95    0.38

──────────
```

**Figure 2  Summary output for example data analysis with Gaussian data (guinea pig analysis).**

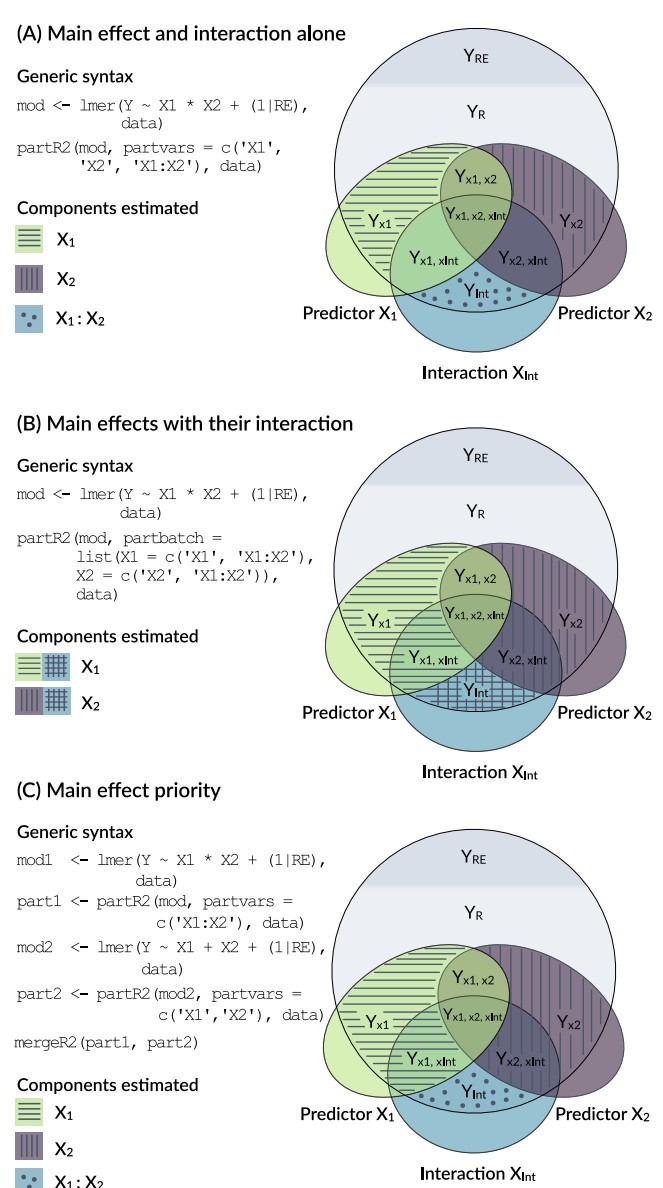

**Figure 3 Conceptual framework for dealing with interactions.** An interaction is the product of two main effects and thus often correlated with each of the main effects. The figure shows three options for estimating the part $R^2$ for main effects that are involved in an interaction.

```
partR2(mod, partvars=c("Rank"), partbatch=list(Time=batch, ''Time:Rank''=
paste0(batch, ":Rank")), nboot=100)
```

This, however, is only one way of dealing with interactions (Option A in Fig. 3). It represents the variance uniquely explained by main effects even in the presence of an interaction. Since interactions are the products of main effects, interaction terms are typically correlated with main effects and the part $R^2$ calculated above might not represent a biologically relevant quantity. There are two alternative ways of how to deal

with interactions. Both are possible in `partR2`, but since requirements differ between applications, we do not implement one with priority.

One way to think about variance explained by main effects and their interactions is to pool the variance explained by a main effect with the variance explained by interactions that the term is involved in (Option B in Fig. 3). In the guinea pig example, for instance, *Rank* might be considered important either as a main effect or in interaction with time and we might want to estimate the total effect of *Rank*. This can be done for the guinea pig dataset by using `partbatch`:

```
mod <- lmer(Testo ~Time * Rank + (1|MaleID), data=GuineaPigs)
partR2(mod, partbatch = list(Time=c("Time", "Time:Rank"), Rank=c("Rank",
"Time:Rank")), nboot=100)
```

A third, which we think usually preferable option is to prioritize main effects by assigning the proportion of variance that is explained by a main effect together with the variance jointly explained with its interaction to the main effect (Option C in Fig. 3). This implies that part $R^2$ for a main effect is estimated when its own interaction is excluded from the model (`mod1` and `part1` below). The variance explained by the interaction is then estimated in a separate model (`mod2` and `part2` below). We have implemented a helper function `mergeR2` that allows to merge two `partR2` runs.

```
mod1 <- lmer(Testo ~Time * Rank + (1|MaleID), data=GuineaPigs)
part1 <- partR2(mod1, partvars = c("Time:Rank"), nboot=100)
mod2 <- lmer(Testo ~Time + Rank + (1|MaleID), data=GuineaPigs)
part2 <- partR2(mod2, partvars = c("Time", "Rank"), nboot=100)
mergeR2(part1, part2)
```

All these results can be viewed by `print`, `summary` and plotted by `forestplot`. It is important to bear in mind the differences in the interpretation as illustrated in Fig. 3.

## An example with proportion data

As an example for proportion data, we analyze a dataset on spatial variation in color morph ratios in a color-polymorphic species of grasshopper. Individuals of this species occur either in a green or a brown color variant and the dataset contains counts of brown and green individuals (separated for females and males) from 42 sites sampled in the field (*Dieker et al., 2018*). Site identity will be fitted as a random effect. As covariates the dataset contains a range of Bioclim variable that describe various aspects of ecologically relevant climatic conditions (*Karger et al., 2017*). The aim is to identify the climatic conditions that favour one or the other colour variant.

We first load the grasshopper dataset. We standardise all Bioclim variables using the `scale` function and add an observation-level counter that will be used as an observation-level random effect (OLRE) to account for overdispersion (*Harrison, 2014*).

```
data(Grasshoppers)
for (i in which(substr(colnames(Grasshoppers),1,3)=="Bio"))
{Grasshoppers[,i] <- scale(Grasshoppers[,i])
}
Grasshoppers$OLRE <- 1:nrow(Grasshoppers)
```

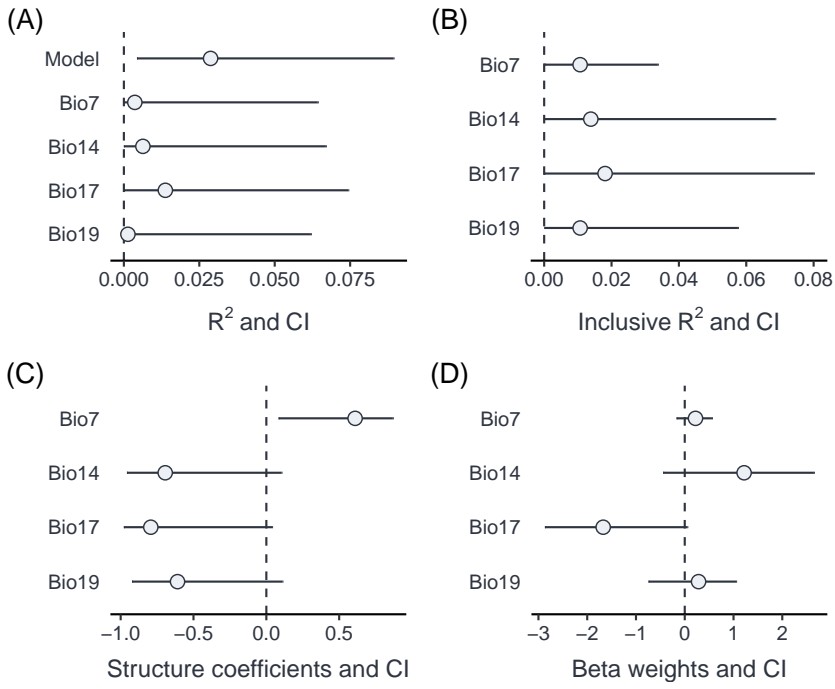

**Figure 4** Comparison of part $R^2$ for individual predictors (A), inclusive $R^2$ (B), structure coefficients (C) and beta weights (D) for an example dataset with proportion data from grasshoppers.

We first fit a GLMM with binomial error structure and logit link using the `glmer` function from the `lme4` package (*Bates et al., 2015*). A previous analysis has shown that the first principle component of the Bioclim data explains a small, but significant part of variation in morph ratios (*Dieker et al., 2018*). For illustration, we use the four Bioclim variables that show a loading of more than 0.30 on the first principle component.

```
mod <- glmer(cbind(nGreen, nBrown) ~ Bio7 + Bio14 + Bio17 + Bio19 +
(1|SiteID) + (1|OLRE), data=Grasshoppers, family="binomial")
res <- partR2(mod, partvars=c("Bio7", "Bio14", "Bio17", "Bio19"),
max_level=1, nboot=100)
```

The `summary` output informs us (at the bottom) that there have been warnings in the bootstrapping processes. This is not unusual since bootstrapping frequently generates data, for which one of the parameters is estimated at the boundary (in particular if one of the variance components is very small). The results can be visualised using the `forestplot` function (Fig. 4). Plotting is based on `ggplot2` (Wickham, 2016), and multiple forest plots can easily be assembled using the `patchwork` package (*Pedersen, 2020*). Forest plots show the effect sizes graphically and can be set to either show part $R^2$ when `type = "R2"` (the default), inclusive $R^2$ when `type = "IR2"`, structure coefficients when `type = "SC"`, and beta weights (standardized model estimates) with `type = "BW"`.

```
p1 <- forestplot(res, type = "R2")
p2 <- forestplot(res, type = "IR2")
p3 <- forestplot(res, type = "SC")
```

```
    p4 <- forestplot(res, type = "BW")
    library(patchwork)
    (p1 + p2) / (p3 + p4) +
plot_annotation(tag_levels = "A", tag_prefix= "(", tag_suffix = ")")
```

A comparison of part $R^2$, inclusive $R^2$, structure coefficients beta weights shows the different insights that can be gained from these different summaries of the model fit (Fig. 3). In this case, three of the Bioclim variables (*Bio14*, *Bio17*, *Bio19*) are highly positively correlated ($r \geq 0.93$), while a fourth one (*Bio7*) is moderately negatively correlated to all three of them ($r \leq -0.63$). Part $R^2$ are thus low, because none of the parameters uniquely explains a large share of the variance. *Bio17* seems to be the best predictor of morph ratios, with the largest (negative) beta weight, largest part $R^2$, largest structure coefficients and largest inclusive $R^2$. Beta weights for the two positively correlated (but slightly weaker) predictors, *Bio14* and *Bio19*, switch sign as is not unusual for collinear predictors. This means that after accounting for the effect of *Bio17*, they contribute positively to prediction. However, structure coefficients show that both variables load negatively on the linear predictor, as does *Bio17*.

## Challenges

Using transformation or functions in the formula argument can lead to issues with matching the terms of the model with the `partvars` argument of `partR2`. It is therefore important that the names in `partvars` match exactly the terms in the `merMod` object. However, any complications are easily circumvented by implementing the transformations before fitting the model and storing them in the data frame used in the analysis. It is also worth to be aware that unusual names may cause complications and renaming can offer an easy solution.

We have repeatedly seen model outputs where the point estimate does not fall within the confidence interval. This might seem like in the bug in the package, but in our experience usually indicates issues with the data and/or the model. In fact, parametric bootstrapping can be seen as a limited form of posterior predictive model checks (*Gelman & Hill, 2006*). If generating new data from the fitted model (as done with parametric bootstrapping) results in data that are dissimilar to the original data, then the model is probably not a good fit to the data.

Bootstrap iterations can sometimes yield slightly negative estimates of part $R^2$, in particular if the variance explained by a predictor is low. These negative estimates happen in mixed-effects models, because estimates of random-effect variance might change when a predictor is removed and this can lead to a slight decrease in the residual variance, and hence a proportional increase in $R^2$ (*Rights & Sterba, 2019*). By default, `partR2` sets negative $R^2$ values to 0, but this can be changed by setting `allow_neg_r2` to TRUE. It also happens that inclusive $R^2$ is estimated slightly lower than part $R^2$ when the contribution of a particular predictor is very large. We consider both cases as sampling error that should serve as a reminder that variance components are estimated with relatively large uncertainly and minor differences should not be over-interpreted.

A warning needs to be added for the estimation of $R^2$ (and, in fact, also repeatability $R$) from small datasets. In particular if the number of levels of random effect is low, variance components might be slightly overestimated ($Xu$, $2003$). This issue applies similarly to the variance explained by fixed effects, in particular if the number of predictors is large relative to the number of data points.

### Code and data availability

The current stable version of partR2 can be downloaded from CRAN (https://cran.r-project.org/web/packages/partR2/index.html) and the development version can be obtained from GitHub (https://github.com/mastoffel/partR2). The data used in the examples is part of the package.

## ACKNOWLEDGEMENTS

We thank Alexandra Mutwill for providing the guinea pig data.

### Funding

Martin A. Stoffel and Holger Schielzeth were supported by the German Research Foundation (DFG) as part of the SFB TRR 212 (NC3) (funding INST 215/543-1, 396782608). Shinichi Nakagawa was supported by the ARC Discovery Project grant (DP180100818). The funders had no role in study design, data collection and analysis, decision to publish, or preparation of the manuscript.

### Grant Disclosures

The following grant information was disclosed by the authors:
German Research Foundation (DFG): INST 215/543-1, 396782608.
ARC Discovery Project: DP180100818.

### Competing Interests

The authors declare there are no competing interests.

### Author Contributions

- Martin A. Stoffel performed the experiments, analyzed the data, prepared figures and/or tables, authored or reviewed drafts of the paper, wrote the R package, and approved the final draft.
- Shinichi Nakagawa conceived and designed the experiments, performed the experiments, analyzed the data, authored or reviewed drafts of the paper, and approved the final draft.
- Holger Schielzeth conceived and designed the experiments, performed the experiments, analyzed the data, prepared figures and/or tables, authored or reviewed drafts of the paper, and approved the final draft.

## Animal Ethics

The following information was supplied relating to ethical approvals (i.e., approving body and any reference numbers):

The Landesamt für Natur, Umwelt und Verbraucherschutz Nordrhein-Westfalen "LANUV NRW" (Germany) approved this research (reference number: 84-02.04.2015.A439).

## Data Availability

The partR2 package and vignette are available at CRAN (https://cran.r-project.org/web/packages/partR2/index.html) and the development version is available at GitHub (https://github.com/mastoffel/partR2). All data used in the article are part of the package.

## Supplemental Information

Supplemental information for this article can be found online at http://dx.doi.org/10.7717/peerj.11414#supplemental-information.

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
