# Peer review of "partR2: partitioning R2 in generalized linear mixed models"

_PeerJ, doi:10.7717/peerj.11414_

## Round 0.1 · original submission · Major Revisions

The reviewers have provided critical comments regarding the mathematical and statistical model. Please respond and revise you manuscript accordingly and I will send the revised version out for re-evaluation.

Reviewer 1 ·

Basic reporting

This manuscript is well-written in a professional way.

Experimental design

The presentation is well-thought and comprehensive.

Validity of the findings

This manuscript presents a useful R package to the research community.

Additional comments

The authors introduce partR2, a versatile package for estimating part R2, inclusive R2, structure coefficients and beta weights from mixed-effects models. They also show how to use partR2 with real example datasets for Gaussian and binomial GLMMs. The manuscript is well-written and the presentation is well-thought and comprehensive.

My only comment is that partR2 can be explained in more detail in the text. It is an important and subtle concept. It would be nice if it can be defined in the text using the notations in Figure 1 so the reader does not need to read Nimon et al. (2008).

Reviewer 2 ·

Basic reporting

In their paper Stoffel et al. describe a potential framework for calculating semi-partial R^2 for GLMMs. Such a contribution would be ground-breaking and very exciting. Some features of the author's function partR2 appear useful. However, I am not convinced that the author's overall approach, in the context of GLMMs, is valid. I have five main concerns:

1) The definition given for R^2 is confusing and non-rigorous. The coefficient of determination is the proportion of the variance in the dependent variable that is predictable from the independent variable(s).

2) The paper introduces an algorithm for calculating semi-partial R^2, however most scientists are much more familiar with partial R^2. Only a brief mention of partial R^2 is given on lines 90-93. A concise comparison of these methods, along with a motivational paragraph concerning the undervalued importance semi-partial R^2 would be useful.

3) The paper lacks mathematical rigor throughout. Only one inline formula is given. Specifically, on line 66 the authors confusingly give the equation for the mean function of a linear model in the context of a sentence describing structure coefficients. Lacking and badly needed are explicit formulae for semi-partial coefficients of determination and structure coefficients, along with mathematical frameworks for general linear mixed models, generalized linear models and generalized linear mixed models.

4) Importantly, it is unclear how isolated correlation coefficients (structure coefficients) squared or otherwise would provide insight concerning partial and semi-partial R^2 in a GLMM.

5) The background and literature cited is insufficient. For instance, missing from the list of R functions and packages that calculate partial and semi-partial R^2 is the library ppcor which has functions that calculate both of these entities, along with performing null hypothesis tests. More troubling, the authors claim that their framework for semi-partial R^2 for GLMMs is based on R^2. However quantifying explained variability in generalized linear models, including those with binomial errors, is a difficult and contentious subject with a large body of literature. None of this work is considered by the authors.

Experimental design

Not applicable here

Validity of the findings

Insufficient background is provided to allow confirmation of the validity of the author's approach.

Reviewer 3 ·

Basic reporting

No comment

Experimental design

Not applicable

Validity of the findings

No comment

Additional comments

I have reviewed the referenced manuscript and find it provides a nice addition to the literature. The manuscript followed the R code well and the contained data files made it easy to replicate the analyses presented. Please find below opportunities to improve the accessibility of the material presented:
1. I think more explanation needs to be provided including formulas for each statistic calculated. I did find some information by using help(partR2) and in the vignettes, but it was not complete. In particular, what is the formula for R2 of which part R2 is based?
2. I am unfamiliar with the term inclusive R2 and wonder if more information could be provided to support how and why it is being calculated.
3. Communality should be commonality on line 92.
4. Line 80 - I disagree that the analysis presented is similar to commonality analysis.

---

## Round 0.2 · Minor Revisions

Thank you for making the revisions. The reviewers have raised some further concerns that need to be addressed.

Reviewer 2 ·

Basic reporting

The paper is dramatically improved with the inclusion of clarifying equations and responses to reviewers. However, a few issues still exist that I have described in an attachment.

Experimental design

Not applicable

Validity of the findings

These seem reasonable now given additional background.

Annotated reviews are not available for download in order to protect the identity of reviewers who chose to remain anonymous.

Reviewer 3 ·

Basic reporting

no comment

Experimental design

no comment

Validity of the findings

no comment

Additional comments

partR2: Partitioning R2 in generalized linear mixed models
I have reviewed the revised version of the referenced manuscript and appreciate how the authors attended to the reviewer comments.
1. Thank you for the definition of inclusive R2 as presented in the manuscript. I am not clear however why a new term is needed is explain the relationship between a criterion and a predictor as multiplying the squared structure coefficient by Multiple R2 always results in the squared validity coefficient or zero-order correlation between the predictor and criterion. This can be seen by examining the formula for structure coefficients (see for example Nathans et al., 2012).
2. Communality should be commonality.
3. It is not clear how the analyses is conceptually related to commonality analysis.
4. Line 80 - I disagree that the analysis presented is similar to commonality analysis.

---

## Round 0.3 · accepted · Accept

The current form of the manuscript is satisfactory and I am happy to accept it for publication.

Reviewer 2 ·

Basic reporting

The authors have addressed my comments adequately.

Experimental design

Not applicable

Validity of the findings

The findings seem reasonable now given additional background.

Reviewer 3 ·

Basic reporting

No comment

Experimental design

No comment

Validity of the findings

No comment

Additional comments

The authors have addressed all previous comments. I appreciate very much the information provided in Figure 1.